 eLIFE

# Distinct gating mechanisms revealed by the structures of a multi-ligand gated K+ channel

Chunguang Kong[1], Weizhong Zeng[1,2†], Sheng Ye[3†], Liping Chen[1,2†], David Bryant Sauer[1], Yeeling Lam[1], Mehabaw Getahun Derebe[1], Youxing Jiang[1,2*]

[1]Department of Physiology, University of Texas Southwestern Medical Center, Dallas, United States; [2]Howard Hughes Medical Institute, University of Texas Southwestern Medical Center, Dallas, United States; [3]Life Sciences Institute, Zhejiang University, Hangzhou, China

**Abstract** The gating ring-forming RCK domain regulates channel gating in response to various cellular chemical stimuli in eukaryotic Slo channel families and the majority of ligand-gated prokaryotic K+ channels and transporters. Here we present structural and functional studies of a dual RCK-containing, multi-ligand gated K+ channel from *Geobacter sulfurreducens*, named GsuK. We demonstrate that ADP and NAD+ activate the GsuK channel, whereas Ca2+ serves as an allosteric inhibitor. Multiple crystal structures elucidate the structural basis of multi-ligand gating in GsuK, and also reveal a unique ion conduction pore with segmented inner helices. Structural comparison leads us to propose a novel pore opening mechanics that is distinct from other K+ channels.

*For correspondence:
youxing.jiang@utsouthwestern.edu

†These authors contributed equally to this work

**Competing interests:** The authors have declared that no competing interests exist

**Reviewing editor**: Richard Aldrich, The University of Texas at Austin, United States

## Introduction

Ligand-gated K+ channels open and close in response to various cellular chemical stimuli. The majority of prokaryotic ligand-gated K+ channels, as well as eukaryotic Slo channel families (Slo1 or BK, Slo2 and Slo3) (*Salkoff et al., 2006*), have one or two copies of a conserved C-terminal intracellular ligand-binding RCK (regulating the conductance of K+) domain (*Jiang et al., 2001*; *Jiang et al., 2002a*; *Kuo et al., 2005*) (*Figure 1A*). RCK domains are also ubiquitously distributed in the bacterial K+ uptake (Trk or Ktr systems) (*Schlösser et al., 1993*; *Nakamura et al., 1998*) and efflux machinery (Kef systems) (*Bakker et al., 1987*; *Munro et al., 1991*). The wide distribution of RCK domains in K+ channels and transporters highlights their importance in regulating K+ transport across the cell membrane.

RCK domains associate as a dimer, (*Jiang et al., 2001*; *Roosild et al., 2002*; *Dong et al., 2005*), which serves as the basic building block for the quaternary structural assembly in both K+ channels and transporters. As demonstrated in the structure of MthK, a Ca2+-gated K+ channel from *Methanobacterium thermoautotrophicum*, four RCK dimers assemble into an octameric gating ring in a functional channel tetramer (*Jiang et al., 2002a*). The same quaternary complex is also observed in the K+ transporter systems (*Albright et al., 2006*), indicating that a gating ring of eight RCK domains is the functional assembly for both channels and transporters. Most prokaryotic RCK-containing K+ channels have only one copy of the RCK domain on each subunit and the formation of an octameric gating ring requires the co-expression of their cytosolic RCK domain via an alternative internal translation start site on the same gene (*Jiang et al., 2002a*). The eukaryotic Slo K+ channel families and a subset of prokaryotic K+ channels already contain two tandem RCK domains on each subunit and therefore the gating ring assembly in these channels no longer requires the co-expression of an isolated RCK domain (*Figure 1B,C*). The formation of the gating ring provides a platform for diverse allosteric ligand regulation among

**eLife digest** Most cells are surrounded by a semipermeable membrane, and although this membrane allows very few molecules to pass through it, cells can use transmembrane proteins to overcome this barrier. Some of these proteins import glucose, amino acids and other nutrients into the cell, while others transport ions into or out of the cell. Ion transport across the cell membrane is essential for a wide variety of biological processes, including signal transduction and the generation of electrical impulses in nerve cells.

The pores that allow ions to travel through the cell membrane are known as ion channels, and most channels allow only one type of ion—usually sodium, calcium or potassium ($K^+$) ions—to pass through them. There are many different types of ion channels and they are classified according to the type of ion they allow to pass through them, and by the gating mechanism that is used to open and close the channel. For example, ligand-gated $K^+$ channels facilitate the passage of potassium ions and are opened and closed by ligands binding and unbinding to and from the channel.

Most $K^+$ channels are made up of four identical subunits, and in the majority of ligand-gated $K^+$ channels in prokaryotes, each of these subunits will have one or two ligand-binding RCK domains (where RCK stands for regulating the conductance of $K^+$). This is also true for some $K^+$ channels in eukaryotes. While it is known that RCK domains are responsible for regulating the transport of potassium ions across the cell membranes of diverse organisms, little is known about the structure or gating mechanisms of $K^+$ channels that are gated by more than one ligand.

Kong et al. have studied a ligand-gated $K^+$ channel called GsuK that has two RCK domains per subunit and is found in the bacterium *G. sulfurreducens*. They found that the opening process was mediated by a ligand that contains adenine, such as $NAD^+$ or ADP, and the channel was closed by the presence of calcium ions. And by determining multiple crystal structures, Kong et al. were able to understand, from a structural point of view, how these ligands regulate this channel, and to propose a gating mechanism that is distinct from the mechanisms that are known to control other potassium channels.

RCK-containing channels, and the gating rings of some channels are susceptible to multiple cellular stimuli. For example, the RCK domains of BK channels have multiple divalent cation ($Ca^{2+}$ and $Mg^{2+}$) binding sites for channel activation and can be modulated by phosphorylation and heme binding, and so on (*Zhang et al., 2001*; *Shi et al., 2002*; *Xia et al., 2002*; *Bao et al., 2004*; *Zeng et al., 2005*; *Hou et al., 2009*).

The structural information of RCK-containing channels has been limited to the low resolution single-RCK MthK channel in a $Ca^{2+}$-bound, open conformation (*Jiang et al., 2002a*) or the isolated cytosolic RCK domains from other $K^+$ channels and transporters (*Roosild et al., 2002*; *Albright et al., 2006*; *Ye et al., 2006*; *Wu et al., 2010*; *Yuan et al., 2010*; *Yuan et al., 2011*). Due to the resolution limit and poor crystal packing, the ion conduction pore of MthK was poorly defined and the linkers between the pore and the gating ring, which are essential for coupling the gating ring conformational change to the pore opening and closing, could not be resolved in the structure. In this study, we present structural and functional studies of a novel a two-transmembrane, RCK-regulated $K^+$ channel, GsuK, named after the bacterium *Geobacter sulfurreducens* from which this **K**$^+$ channel was cloned. Each GsuK subunit contains two tandem RCK domains, reminiscent of Slo $K^+$ channels. We demonstrate that GsuK is a nucleotide-activated and $Ca^{2+}$-deactivated $K^+$ channel and reveal the structural mechanism of multi-ligand gating of this double-RCK $K^+$ channel and a distinct pore opening mechanics.

## Results

### Structure of the intracellular ligand-binding gating ring of GsuK

Our study of GsuK started with the structure determination of its two tandem RCK domains, labeled as RCK1 and RCK2. The two intracellular ligand binding RCK domains form a bi-lobed structure equivalent to the MthK RCK dimer; each lobe consists of the N-terminal two-thirds of the RCK domain

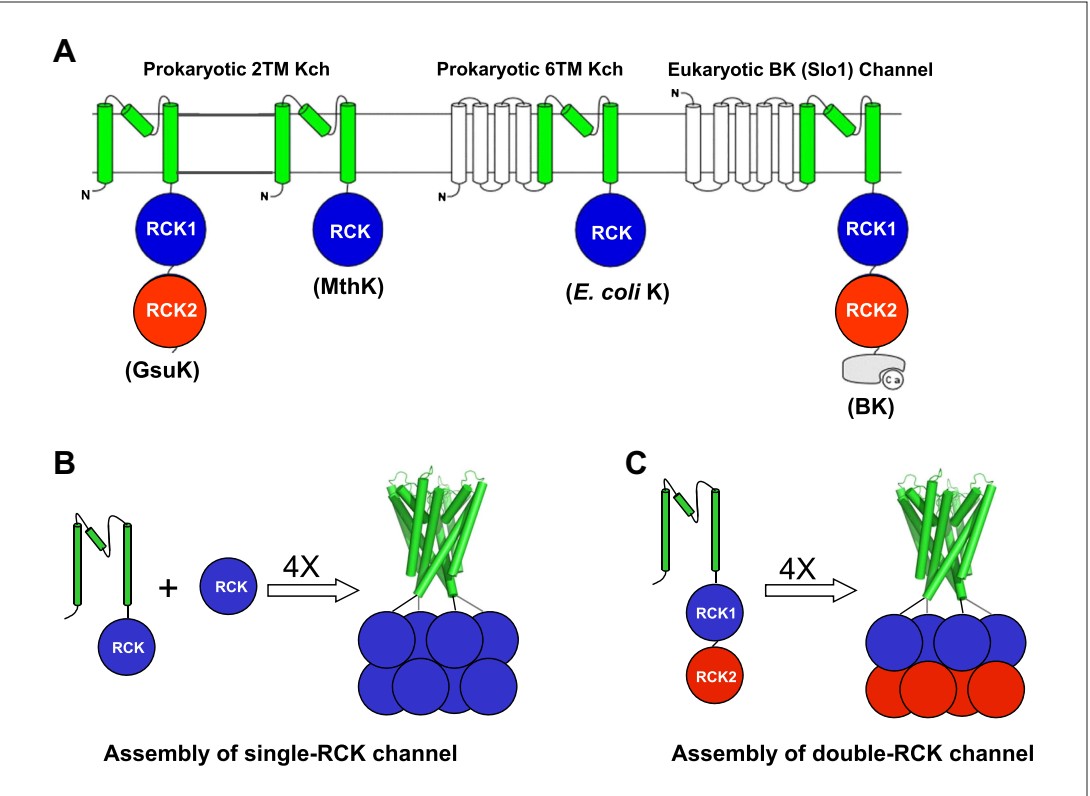

**Figure 1**. RCK-Regulated K+ Channels. (**A**) Topology of RCK-regulated K+ channels. (**B**) Functional assembly of single RCK-containing channels such as MthK. (**C**) Functional assembly of double RCK-containing channels such as GsuK in this study.

(βA to αF) and adopts a Rossmann-fold (**Rossmann et al., 1974**) (**Figures 2 and 3**). While the secondary structural elements of the C-terminal subdomains are similar between GsuK and MthK, their tertiary structural arrangements are quite distinct. In MthK, the N-terminal lobes and the C-terminal subdomains of the RCK dimer are connected by interlocking helix-turn-helix motifs (αF-turn-αG), which provide extensive dimerization interactions at the so-called flexible interface (**Figure 3B**). In GsuK, the equivalent αG helix is absent in RCK1 and becomes a shorter helix with a different orientation in RCK2, resulting in swapped and loosely packed C-terminal subdomains (**Figure 3A**). Four GsuK intracellular subunits assemble into a gating ring containing eight RCK domains through inter-subunit interactions at the assembly interfaces (**Figure 4A,B**), with the N-terminal Rossmann-folded lobe of each RCK forming the core of the gating ring and the C-terminal subdomain loosely associating with the core of the gating ring on the periphery.

As the GsuK gating ring is formed by two different sets of RCK domains, its top and bottom halves are not twofold symmetrical, as seen in the MthK gating ring. The pore-connecting top half of the GsuK gating ring is in a contracted form similar to the closed MthK whereas the bottom half is more expanded (**Figure 4C**), suggesting that the structure likely represents a closed conformation. Furthermore, each subunit also contains a small fragment of the pore-lining inner helix at its N-terminus, which forms a short four-helix bundle atop the center of the gating ring and creates a constriction point at the intracellular end of the pore that would occlude the passage of hydrated K+ ions (**Figure 4A,D**). The same closed gate is also observed in the structures of the full-length channel as discussed later. The helix bundle is tethered to RCK1 by linkers in an extended configuration, ensuring a tight coupling between the gating ring conformational change and pore opening at the intracellular gate (**Figure 4D**). Despite low sequence similarity, the position and structural features of this linker are similar to that observed in the BK channel gating ring (**Wu et al., 2010**) (**Figure 4D**).

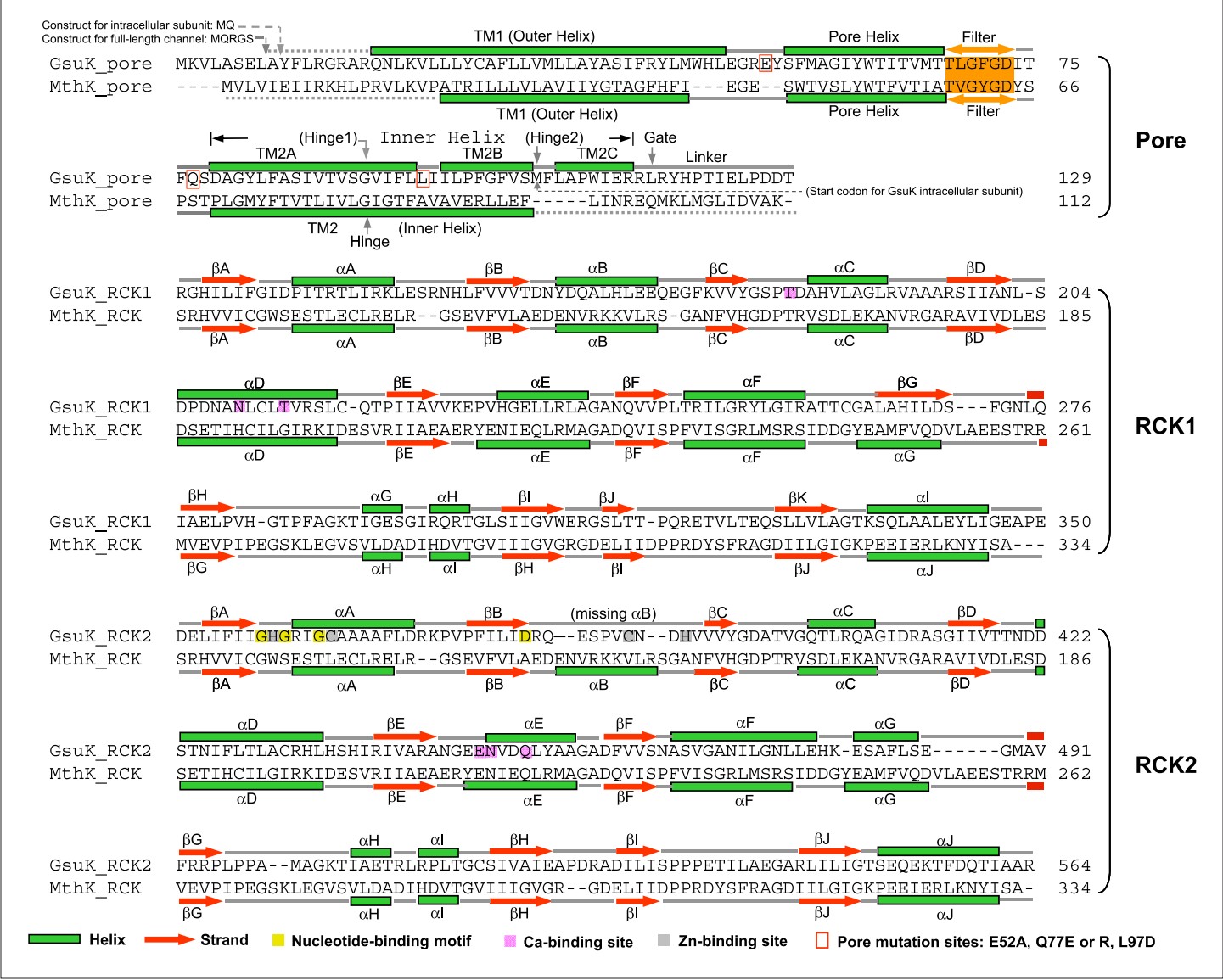

**Figure 2**. Sequence and secondary structure comparison between GsuK and MthK. For comparative purposes, the secondary structural elements of each GsuK RCK domain are labeled following the same nomenclature used for MthK. A duplicate copy of MthK RCK is used in the alignment with GsuK RCK2.

## Multiple ligand binding in GsuK gating ring

The RCK2 domain of GsuK contains the conserved GxGxxG…D/E sequence motif for nucleotide binding in Rossmann-folded protein (**Bellamacina, 1996**) (**Figure 2**). Indeed, electron density modeled as AMP was observed at the predicted nucleotide binding site in the GsuK gating ring structure (**Figure 5A**). As no nucleotides were added during protein purification or crystallization, the bound nucleotide is likely from the *E. coli* cells used for protein expression. Although modeled as AMP, the electron density could actually be from other adenine-containing nucleotide whose AMP moiety is well structured while the rest is mobile. This is the case in the structure of the nucleotide binding RCK domain from a $K^+$ transporter, in which only the AMP portion of a bound $NAD^+$ can be defined (**Roosild et al., 2002**; **Albright et al., 2006**). Supporting this hypothesis, our functional characterization using the $^{86}Rb$ flux assay demonstrated that ADP and $NAD^+$ are the likely ligands for GsuK as discussed below.

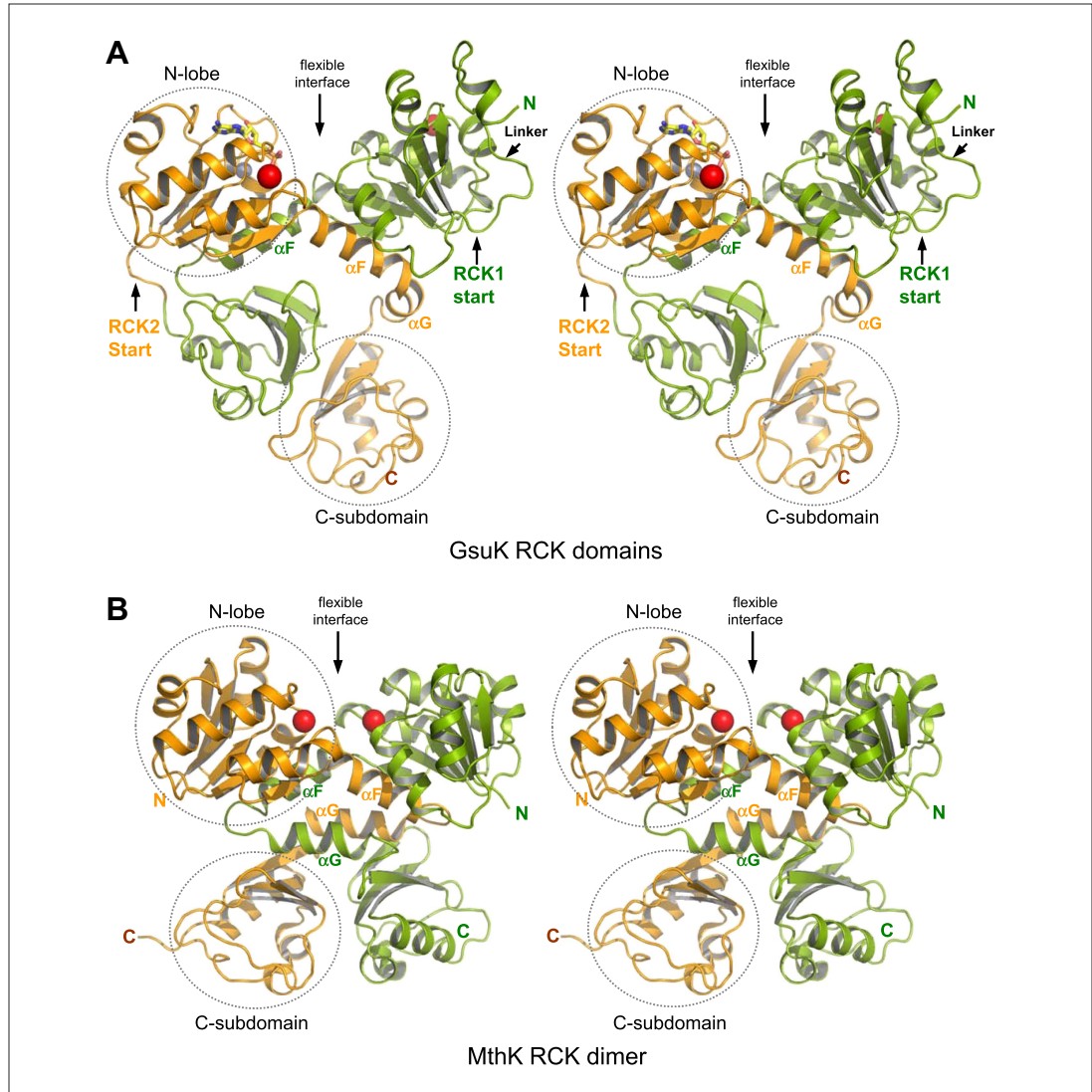

**Figure 3**. Structure of the GsuK intracellular subunit. (**A**) Stereoviews of GsuK intracellular subunit. RCK1 and RCK2 are colored green and orange, respectively. $Ca^{2+}$ and $Zn^{2+}$ ions are shown as red and silver spheres, respectively. The same color representations are used in all figures. (**B**) Stereoviews of MthK RCK dimer. The N-terminal lobes and the C-terminal subdomains are circled in RCK2 of GsuK and in one of the RCK subunits of MthK.

Two bound metal ions were observed in each GsuK subunit. One is identified as $Zn^{2+}$ as it is chelated by His (His359 and His391) and Cys, (Cys364 and Cys388) in RCK2 (*Figure 5B*) with ion coordination chemistry and the local structure resembling a zinc-finger motif; fluorescence scanning of the crystal at the synchrotron also confirmed the presence of $Zn^{2+}$ in the crystal. Whether $Zn^{2+}$ plays any functional role is still unclear and warrants further study. The second bound ion, identified as $Ca^{2+}$ based on ligand chemistry and functional assays, is positioned at the inter-subunit assembly interface, a location reminiscent of the $Ca^{2+}$ bowl in BK channel (*Figure 5C,D*). The six oxygen ligands, two of which are backbone carbonyl oxygen atoms, come from Thr183, Asn210 and Thr214 of RCK1 and Glu449, Asn450 and Gln453 of RCK2 from the neighboring subunit (*Figure 5C*). Unlike that in MthK or BK, $Ca^{2+}$ serves as an allosteric inhibitor in GsuK whose binding stabilizes the closed gating ring and deactivates the channel as confirmed by single channel electrophysiology and the full-length channel structure. As the crystallization conditions contain neither $Zn^{2+}$ nor $Ca^{2+}$, both

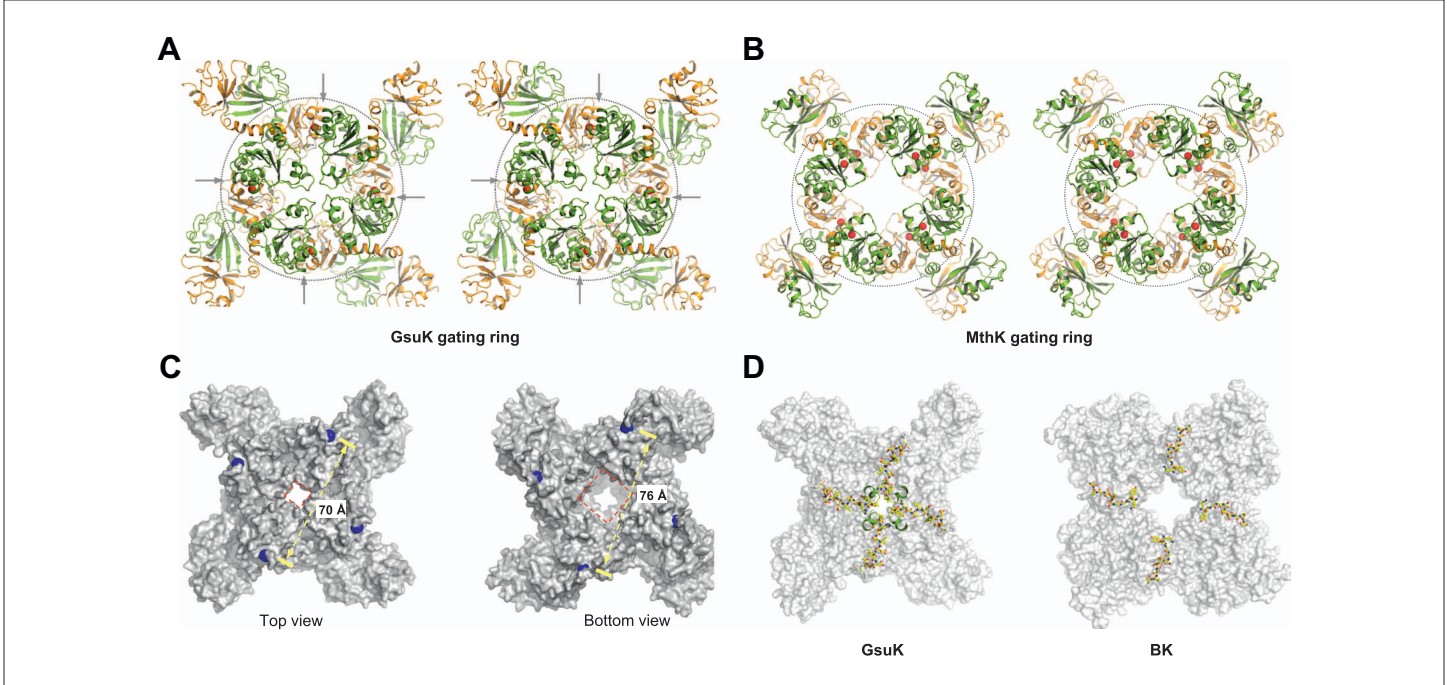

**Figure 4**. Structure of the GsuK gating ring. (**A**) Stereo representation of the GsuK gating ring viewed from the top. Arrows indicate the inter-subunit assembly interface. (**B**) Stereo view of the symmetrical MthK gating ring in the open state. (**C**) Dimension of the GsuK gating ring viewed from top (left) and bottom (right). The diagonal distance is measured between the Ca atoms of Gly131, which is the starting residue of the RCK1. Red square marks the size of the central hole. (**D**) The position of linkers between the gating ring and the ion conduction pore in GsuK (left) and BK channel (right). The linkers are in ball-and-stick representation and the gating rings are shown as surface rendered representation. The short N-terminal four-helix bundle on top of the GsuK gating ring is shown as green ribbons.

ions are likely from the *E. coli* cells or trace amounts of contaminants in the solutions used for protein purification and crystallization.

## Identify the possible ligands for GsuK using flux assay

We utilized the $^{86}$Rb flux assay initially to identify the potential nucleotide ligands for GsuK ('Materials and methods'). In this assay, various nucleotides at a concentration of 1 mM were added individually to GsuK-containing liposomes loaded with high [K$^+$] followed by mixing with the flux buffer containing radioactive $^{86}$Rb. The effect of individual nucleotide on $^{86}$Rb influx into the liposome was monitored by measuring intraliposomal radioactivity levels 5 min after initial mixing. Among those nucleotides tested, guanidine nucleotides had no effect on $^{86}$Rb influx as compared to the control liposomes absent of nucleotide, while ADP and NAD$^+$ led to about a 4–5-fold increase in intraliposomal radioactivity (*Figure 6A*). Interestingly, other adenine-containing nucleotides such as ATP, AMP, or NADH had no obvious effect on $^{86}$Rb influx.

## GsuK is a nucleotide activated, Ca$^{2+}$ deactivated K$^+$ channel

Giant liposome patching was employed to assay the functional properties of both wild-type GsuK and L97D pore mutant, a mutation that enhances the channel conductance and open probability (*Shi et al., 2011*), and also yields better diffracting crystals ('Materials and methods'). In the absence of intracellular Ca$^{2+}$, both the wild-type and mutant channels exhibit high single channel activity with an open probability (P$_o$) of about 0.9 (*Figure 6B–D*). The L97D mutation resulted in a significantly higher single channel conductance than the wild-type channel. Both channels are weakly K$^+$ selective, with a reversal potential of about −40 mV under bi-ionic conditions, equivalent to a permeability ratio (P$_{Na}$/P$_K$) of about 0.2 (*Figure 6D*). The weak K$^+$ selectivity could be partly attributed to the presence of Phe instead of Tyr in the selectivity filter as demonstrated in a recent selectivity

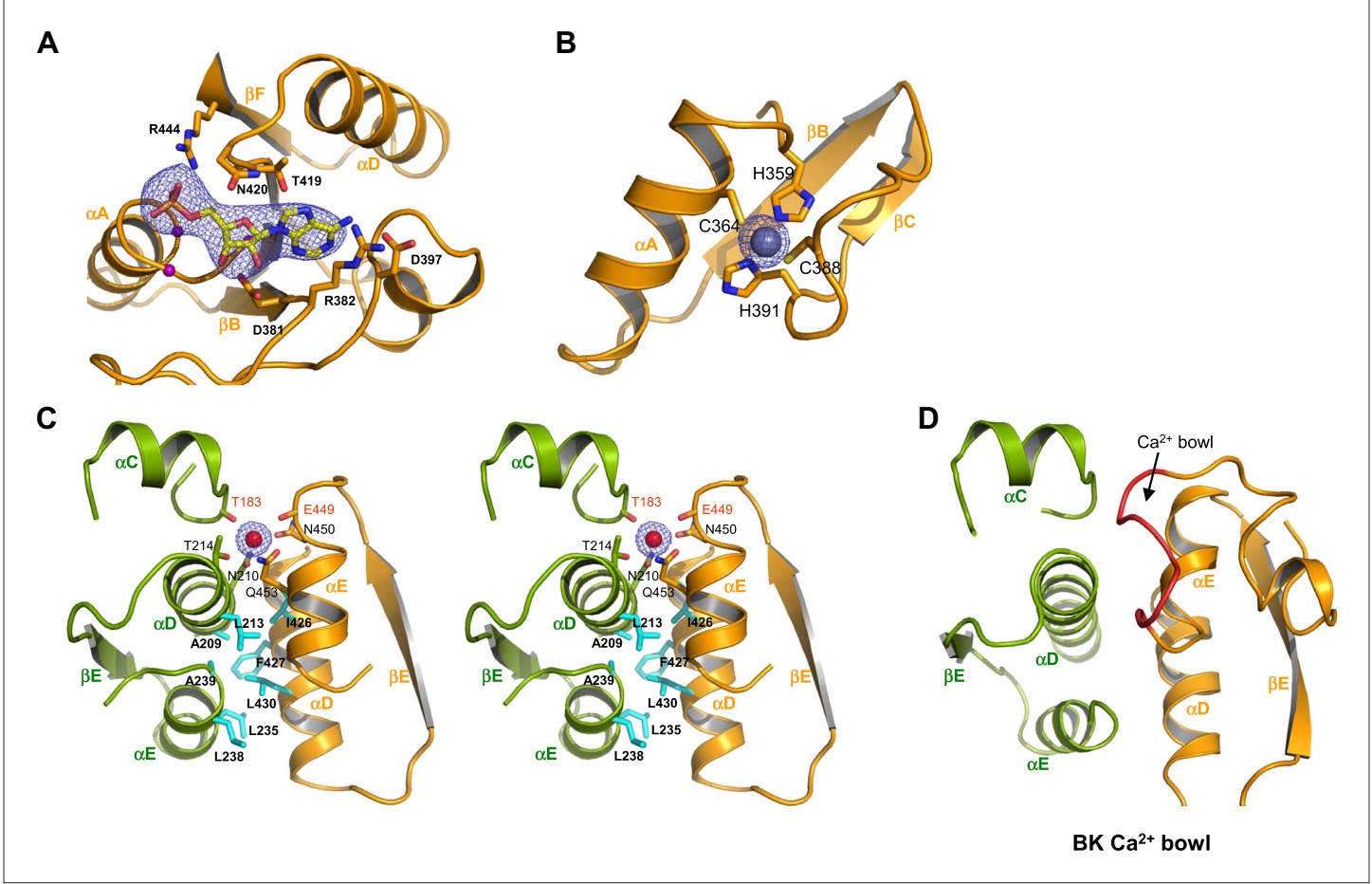

**Figure 5**. Ion and ligand binding in GsuK. (**A**) Structure of the nucleotide-binding site on RCK2. The electron density (blue mesh, contoured at 3σ) from $F_o$–$F_c$ omit map is modeled as AMP. Purple spheres represent the Cα atoms of glycine residues from the conserved nucleotide binding motif. (**B**) Local structure of the $Zn^{2+}$ (silver sphere) binding site on RCK2 with $F_o$–$F_c$ ion omit map (blue mesh) contoured at 9σ. (**C**) Stereoview of the inter-subunit interactions at the assembly interface. Side chains of hydrophobic residues are shown as cyan sticks. Residues that chelate the $Ca^{2+}$ ion with backbone carbonyl oxygen atoms are labeled in red. The electron density of $Ca^{2+}$ (red sphere) from $F_o$–$F_c$ ion omit map is contoured at 5.5σ. (**D**) The inter-subunit interface and $Ca^{2+}$ bowl (red loop) of the BK channel.

study on $K^+$ channel in which replacing Tyr with Phe in the filter results in a decreased $K^+$ selectivity (***Sauer et al., 2011***).

The presence of $Ca^{2+}$ at the intracellular side decreases the channel open probability but not single channel conductance (***Figure 6E***), whereas the presence of $Ca^{2+}$ at the extracellular side has no obvious effect on channel open probability ('Materials and methods'). These results, combined with the observation of $Ca^{2+}$ binding at the gating ring assembly interface, suggest that the reduction in channel activity by $Ca^{2+}$ is a result of a gating rather than pore blocking effect. The wild-type channel exhibits cooperative $Ca^{2+}$ deactivation with a $K_{1/2}$ of 197 μM and Hill coefficient of 2.3. The L97D mutant, on the other hand, mitigates $Ca^{2+}$ deactivation and maintains a fairly high open probability ($P_o$ ~ 0.4) even in the presence of 3 mM $Ca^{2+}$, suggesting that this inner helix mutation favors the pore in an open conformation and thereby weakens the inhibitory gating effect of $Ca^{2+}$. Similar pore opening effect was also observed with equivalent mutation (A88D) in MthK (***Shi et al., 2011***).

The gating effects of nucleotide ligands identified by the flux assay were also assessed on the wild-type channel. Although the gating effect is not as profound as $Ca^{2+}$, both $NAD^+$ and ADP can increase the channel open probability, and their activation effect is more obvious under a $Ca^{2+}$-deactivated state (>300 μM [$Ca^{2+}$]) and less so at lower [$Ca^{2+}$] where the channel are already highly active (***Figure 6F***).

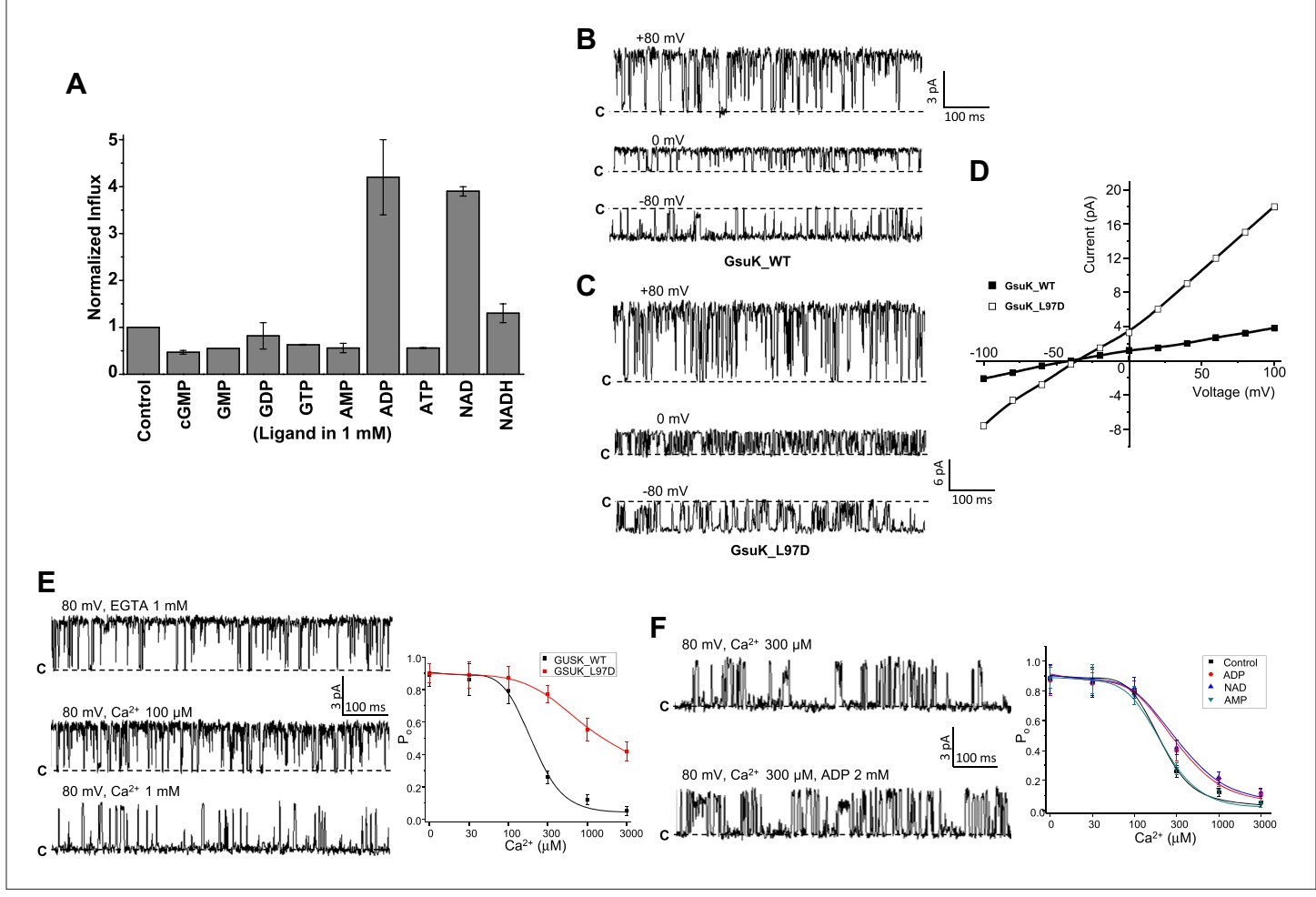

**Figure 6**. Functional analysis of the GsuK channel. (**A**) $^{86}$Rb$^+$ flux assays of GsuK-containing liposomes in the presence of various nucleotides. Data are averages of two measurements and normalized against the control sample without nucleotide. (**B**)–(**D**) Single channel traces and I–V curves of the wild-type channel and L97D mutant. Currents were recorded using giant liposome patch clamping with 150 mM NaCl and 150 mM KCl in the pipette and bath solutions, respectively. (**E**) Sample traces of wild-type channel in the presence and absence of intracellular Ca$^{2+}$ (left) and the plot of [Ca$^{2+}$]-dependent single channel open probability of wild-type GsuK and L97D mutant (right). Both pipette and bath solutions contain symmetrical 150 mM KCl. Data for wild-type channel are fitted to the Hill equation with K$_{1/2}$ of 197 uM and Hill coefficient n = 2.3. Data are mean ± SEM of seven measurements. (**F**) Sample traces of partially deactivated GsuK channel in the presence and absence of 2 mM ADP. Shown on the right is the plot of [Ca$^{2+}$]-dependent single channel open probability of wild-type GsuK in the presence of 2 mM various adenine-containing nucleotide. Both pipette and bath solutions contain symmetrical 150 mM KCl. Data are fitted to the Hill equation with K$_{1/2}$ = 210 μM and n = 2 for AMP, K$_{1/2}$ = 350 μM and n = 1.4 for ADP, and K$_{1/2}$ = 370 μM and n = 1.3 for NAD$^+$. Data are mean ± SEM of five measurements.

Consistent with the flux assay, such an activation effect is not observed with other adenine-containing nucleotides such as AMP.

## Structures of the full-length GsuK channel

Two full-length channel structures, wild-type channel and L97D mutant, were determined at 3.7 Å and 2.6 Å, respectively (*Table 1*). Both channels share a similar overall structure and, therefore, the higher resolution L97D mutant is used here for the description of the overall full-length channel structure (*Figure 7A*). The L97D mutant channel crystals are of the space group *C2* with unit cell dimensions of a = 232.9 Å, b = 111.7 Å, c = 164.1 Å and β = 134.5°, and contain four channel subunits in an asymmetric unit. These subunits do not belong to the same channel tetramer and, instead, are divided into two half channels, which participate in the formation of two channel tetramers with their crystallographic twofold related partners. The gating ring of the full-length channel tetramer adopts a similar

**Table 1.** Data collection and refinement statistics

| Data Collection | Intracellular subunit | Wild type | L97D mutant | L97D mutant /ADP | L97D mutant /NAD |
|---|---|---|---|---|---|
| Space group | I222 | C2 | C2 | C2 | C2 |
| Cell dimensions: a, b, c (Å) | 110.6, 161.7, 310.1 | 235.0, 108.4, 165.8 | 232.9, 111.7, 164.1 | 234.3, 111.4, 164.7 | 232.5, 111.1, 164.6 |
| α, β, γ (°) | 90, 90, 90 | 90, 135.0, 90 | 90, 134.5, 90 | 90, 134.9, 90 | 90, 134.8, 90 |
| Wavelength (Å) | 0.9792 | 1.0332 | 0.9786 | 0.9792 | 0.9792 |
| Resolution (Å) | 50.0 – 3.0 | 50.0 – 3.7 | 50.0 – 2.6 | 50.0 – 2.8 | 50.0 – 3.2 |
| Measured reflections | 309,562 | 123,263 | 303,788 | 225,169 | 141,834 |
| Unique reflections | 55,435 | 27,539 | 87,305 | 72,260 | 47,223 |
| Redundancy* | 5.6 | 4.5 | 3.5 | 3.1 | 3.0 |
| Completeness (%, highest shell) | 99.2 (93.6) | 86.7 (47.4) | 94.7 (57.8) | 96.4 (88.9) | 95.6 (92.5) |
| Mean I/σI (highest shell) | 35.9 (1.8) | 19.0 (1.0) | 18.4 (1.0) | 15.6 (1.0) | 14.7 (1.0) |
| $R_{sym}$ (%, highest shell)† | 6.0 (68.3) | 7.8 (82.9) | 6.4 (51.6) | 7.0 (86.7) | 7.1 (79.2) |
| Refinement | | | | | |
| Resolution (Å) | 50.0 – 3.0 | 50.0 – 3.7 | 30.0 – 2.6 | 50.0 – 2.8 | 50 – 3.2 |
| No. of reflections \|F\|>0 σF | 54,224 | 27,458 | 87,242 | 71,570 | 47,059 |
| R-factor/R-free (%)‡ | 22.8/26.2 | 26.1/29.3 | 20.3/24.9 | 21.3/25.5 | 23.0/27.0 |
| No. of protein atoms | 13,840 | 14,246 | 14,235 | 14,244 | 14,255 |
| No. of solvent atoms | 0 | 0 | 391 | 273 | 50 |
| No. of ions ($K^+$/$Ca^{2+}$/$Zn^{2+}$) | 0/4/4 | 12/4/4 | 12/4/4 | 11/4/4 | 12/4/4 |
| No. of ligands | 4 AMP | 0 | 0 | 4 ADP | 4 NAD |
| Rmsd bond lengths (Å)§ | 0.009 | 0.006 | 0.004 | 0.005 | 0.003 |
| Rmsd bond angles (°) | 1.243 | 1.053 | 0.666 | 0.902 | 0.723 |

*Redundancy = total measurements/unique reflections.

†$R_{sym} = \Sigma|Ii - <Ii>|/\Sigma Ii$, where $<Ii>$ is the average intensity of symmetry equivalent reflections.

‡R factor = $\Sigma|F(obs) - F(cal)|/\Sigma F(obs)$, 5% of the data were used in the $R_{free}$ calculations.

§Rmsd = root-mean-square deviation.

Numbers in parentheses are statistics for highest resolution shell.

structure to the isolated gating ring, indicating a closed conformation. Both $Zn^{2+}$ and $Ca^{2+}$ are present in the full-length channel structure, but no clear density for nucleotide is observed.

The full-length GsuK structure exhibits two major differences in relative position between the gating ring and the pore as compared to MthK. First, GsuK has a long inner helix that is seven residues (about two helical turns) longer than that of MthK, resulting in the attached gating ring being further away from the membrane (*Figure 7B*). Second, GsuK and MthK adopt different relative orientations between the gating ring and the pore (*Figure 7C*). With their pores superimposed, the gating rings of MthK and GsuK have about a 50-degree rotation relative to each other about the central axis. This difference in relative orientation could contribute to the different pore opening mechanics between GsuK and MthK as discussed later.

## Ion conduction pore of GsuK

While having the same structure at the selectivity filter region, the membrane-spanning pore of GsuK has several unique structural features as compared to other $K^+$ channels. First, instead of forming a

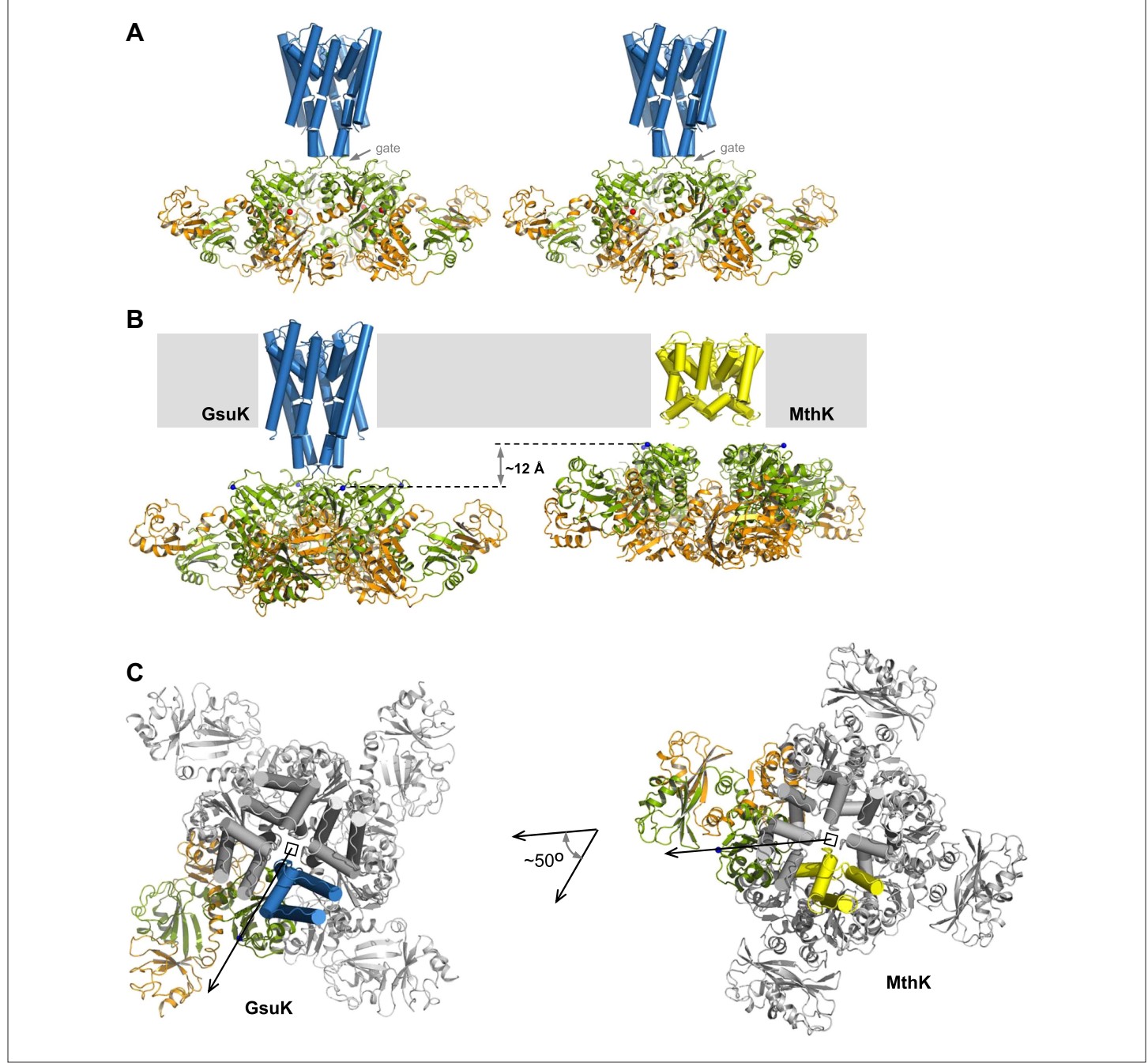

**Figure 7**. Structure of the full-length GsuK channel. (**A**) Stereoview of full-length GsuK channel with L97D mutation. The transmembrane helices are shown as blue cylinders and the gating ring is in ribbon representation with RCK1 in green and RCK2 in orange. Subdomains from the front and back subunits are disordered and absent in the structure. (**B**) Comparison of the translational distances between the gating ring and the membrane-spanning pore in GsuK (left, L97D mutant) and MthK (right). (**C**) Comparison of the relative orientation between the gating ring (ribbon representation) and ion conduction pore (cylinder representation) in GsuK (left) and MthK (right). Only one subunit from each channel is colored. Both channels are superimposed using the pore region and viewed from the extracellular side. Arrows connect the central fourfold axis (square) to the starting residue (Cα atoms of Gly131 in GsuK and Arg114 in MthK) of the first RCK domains, indicating the approximate direction of the linker.

single straight helix, the long inner helix of GsuK is segmented into three parts, labeled TM2A, 2B and 2C, respectively (*Figure 8A*). The break between TM2A and 2B is at a position near the helix-breaking PVP region of Kv channels (*Long et al., 2007*). Second, rather than forming a bundle crossing right below the central cavity as seen in the closed KcsA structure (*Doyle et al., 1998*; *Zhou et al., 2001*),

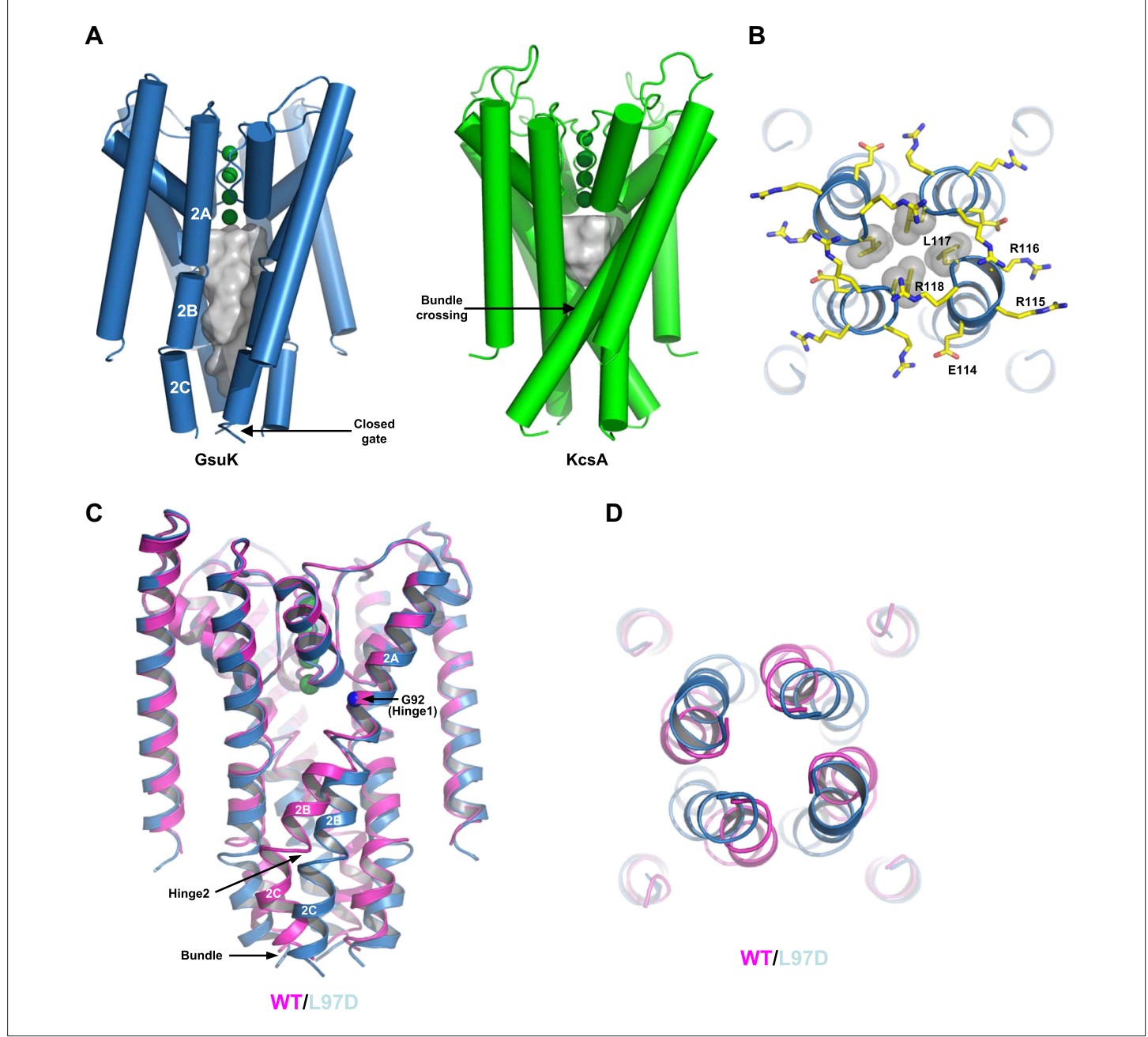

**Figure 8**. The ion conduction pore of GsuK. (**A**) Structural comparison between the GsuK pore and KcsA. K⁺ ions in the filter are shown as green spheres. Grey surface representation illustrates the space of the central cavity. GsuK inner helix (TM2) is segmented into three parts labeled as 2A, 2B and 2C, respectively. (**B**) Zoomed-in view of the GsuK intracellular gate with the surrounding charged residues drawn as sticks. Leu117 side chains are shown in CPK models. (**C**) Superimposition of the ion conduction pores from the wild-type (magenta) and the L97D mutant (blue) channels. (**D**) View of the superimposition from the intracellular side. The intracellular gate remains closed in the L97D mutant.

the four inner helices of GsuK are more parallel to the central pore axis and generate an elongated, water-filled vestibule that spans twice the length of the KcsA cavity (***Figure 8A***). Third, the second inner helix break reorients the TM2C segment towards the central axis, forming a constriction at the very end in a channel tetramer and pinching shut the pore at residue Leu117 (***Figure 8A,B***).

The closed gate is tethered to the RCK gating ring through the extended linkers as was seen in the isolated gating ring structure. A stretch of charged residues, mainly positive ones, are clustered at the

end of the inner helix and cuff around the closed gate (*Figure 8B*). These positively charged residues, also observed in some RCK-containing Slo channels, could potentially participate in channel gating by interacting with lipids similar to the mechanism in inwardly rectified K+ channels (*Hansen et al., 2011*; *Whorton and MacKinnon, 2011*).

Despite the similar gating ring structures, the ion conduction pores of the wild-type and L97D mutant channels exhibit obvious differences along the inner helix. As shown in the pore superimposition (*Figure 8C,D*), the inner helices of the wild-type and mutant channels diverge at the glycine gating hinge (Gly92), but converge at the C-terminal end of the inner helix where the bundle crossing forms. In the L97D mutant, a noticeable helix bend occurs at Gly92 (hinge 1), whereas a sharp turn occurs between TM2B and 2C (hinge 2) in the wild-type channel inner helix.

## ADP and NAD+ binding in GsuK

To reveal the structural basis of nucleotide activation, we also co-crystallized the L97D mutant with ADP and NAD+ and determined the complex structures at 2.8 and 3.2 Å, respectively (*Table 1*). The overall structure of the nucleotide-bound L97D mutant is similar to that of the apo state, with the gating ring in a $Ca^{2+}$-bound, closed conformation.

In the ADP-bound structure, the AMP moiety of the ligand can be unambiguously defined in all four subunits within the asymmetric unit at the same position as was observed in the isolated gating ring structure (*Figure 9A*). The electron density for the β-phosphate group, although not as well resolved as the rest due to higher mobility, points to two possible configurations. In one subunit, the ADP is in an extended *trans* configuration, whereas in the other subunits the ADP is in a *cis* configuration with respect to the phosphoester bond. In the *cis* configuration, the β-phosphate makes a sharp turn and inserts itself into a pocket formed by the loops from βA-to-αA and from βD-to-αD on the second RCK domain (*Figure 9A,B*). The pocket is located at the base of the flexible interface and analogous to the $Ca^{2+}$ binding site in the MthK RCK (*Figure 9C*), suggesting that ADP binding in the *cis* configuration represents an activating state and promotes the gating ring conformational change at the flexible interface similar to $Ca^{2+}$ binding in MthK. The pocket is large enough to accommodate one phosphate group and is in a position only accessible by the β-phosphate, explaining the ligand specificity for ADP but not ATP or AMP.

The NAD+-bound structure has lower ligand occupancy, likely due to a lower ligand affinity in the $Ca^{2+}$-bound closed gating ring. Nevertheless, partially occupied nucleotide is clearly visible in the $F_o$–$F_c$ omit map in two of the subunits (*Figure 10A*). In particular, the ADP and nicotinamide moieties of NAD+ can be properly defined in the electron density map as both are engaged in direct interactions with the protein, whereas the bridging ribose in between is flexible. The position of the nicotinamide group suggests a different activating mechanism between NAD+ and ADP. The NAD+ nicotinamide group is inserted beneath the N-terminal end of the crossover αF helix in RCK1 (*Figure 10A,B*), at the hinge of the flexible interface between βF and αF where the gating ring conformational change occurs (*Ye et al., 2006*). The strategic position and the positive charge suggest that the nicotinamide serves as the activation group of NAD+ and works as a lever whose insertion promotes the hinged motion at the flexible interface towards the open conformation in GsuK. One plausible explanation for the specificity for NAD+ but not NADH is that the tightly-spaced binding pocket permits the insertion of a flat pyridine ring from the nicotinamide of NAD+ but excludes the puckered dihydropyridine ring from NADH.

## Discussion

The structure of GsuK provides an excellent model system for understanding the structural basis of multi-ligand regulation of the RCK gating ring as commonly seen in the eukaryotic Slo channels. $Ca^{2+}$ stabilizes the closed gating ring by binding at the inter-subunit assembly interfaces and deactivates the GsuK channel. $Ca^{2+}$ deactivation has a dominant effect on GsuK gating as the removal of $Ca^{2+}$ increases the channel open probability regardless of the presence or absence of nucleotides. The gating effect of nucleotide binding is less profound than $Ca^{2+}$ as it only moderately enhances the channel open probability under a $Ca^{2+}$-deactivated state. While both NAD+ and ADP anchor their adenine rings at the same site on RCK2 by using the conserved nucleotide binding motif, they utilize different functional groups and activate the channel at different sites. In addition to nucleotide and $Ca^{2+}$, the GsuK RCK2 also has a high affinity site for $Zn^{2+}$ ion whose functional role, if any, is unclear. Despite differences in ligands, there is a convergence of the ligand activation sites between GsuK and

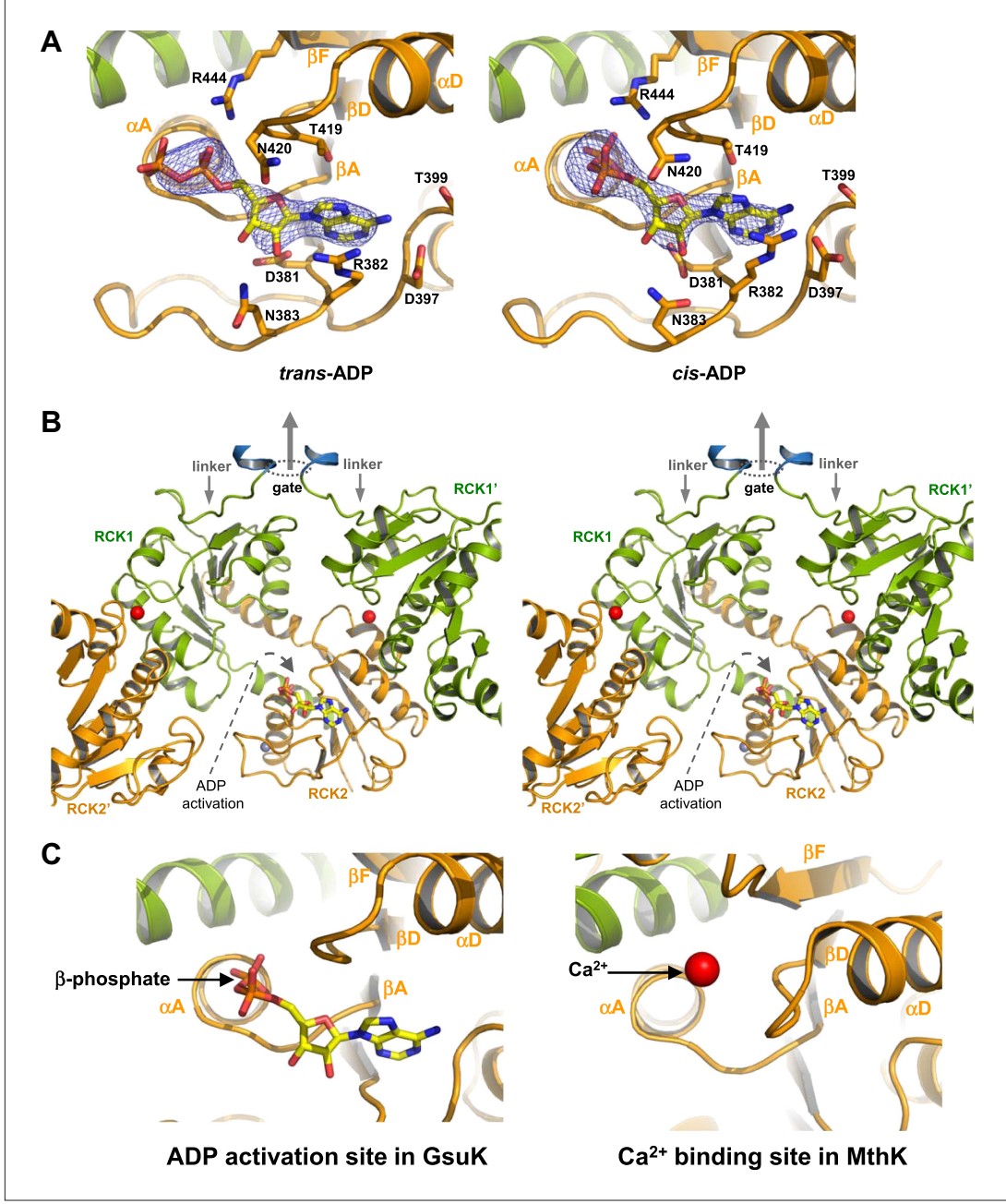

**Figure 9**. ADP binding in GsuK. (**A**) Bound ADP in *trans* (left, non-active) and *cis* (right, active) configurations. The electron density is from $F_o$–$F_c$ omit map contoured at 3.0σ. (**B**) Stereoview of the *cis*-ADP activation site from inside the gating ring. RCK1' and RCK2' are from the neighboring subunits. Dotted oval indicates the location of the gate and the central arrow indicates the direction of the central axis of the pore. (**C**) Local structure comparison between the ADP activation site in GsuK and the $Ca^{2+}$ activation site in MthK.

other RCK-regulated channels. Albeit with opposite gating effect, the $Ca^{2+}$ binding site of GsuK is analogous to the position of the $Ca^{2+}$-bowl in BK (***Wu et al., 2010***; ***Yuan et al., 2010***); and the ADP activation site in GsuK is equivalent to the $Ca^{2+}$ binding site in MthK (***Jiang et al., 2002a***; ***Dong et al., 2005***). In addition, many RCK domains from channels or transporters contain the conserved nucleotide binding motif at the same position, indicating a common site for adenine-containing nucleotide ligand.

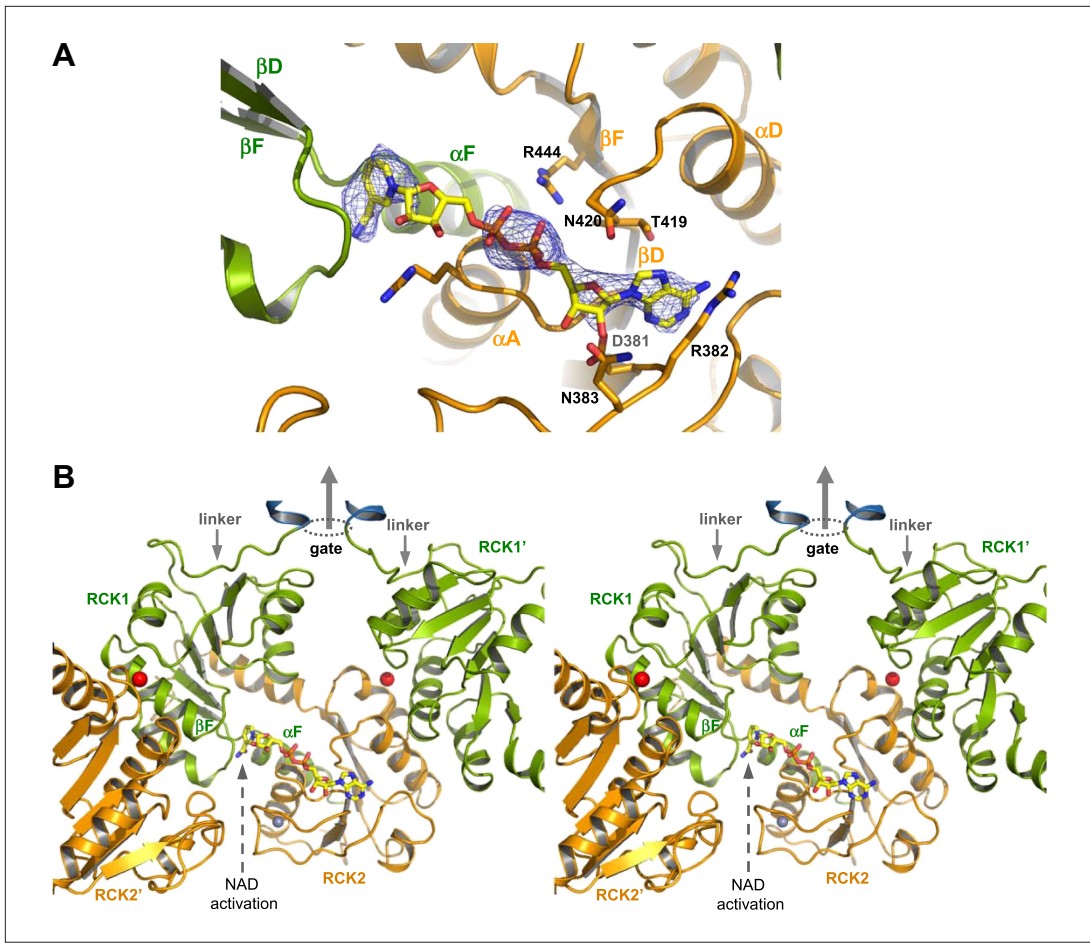

**Figure 10**. NAD+ binding in GsuK. (**A**) NAD+ binding in GsuK with $F_o$–$F_c$ omit map contoured at 3.0σ. (**B**) Stereoview of the NAD+ activation site from inside the gating ring.

GsuK has its intracellular gate at the end of the inner helix, distal from the canonical bundle crossing seen in KcsA, allowing for direct coupling to the gating ring through the extended linker. Although the intracellular gates of both the wild-type and L97D mutant channels are closed in the structures, the structural difference between their ion conduction pores suggests a distinct pore opening mechanism for GsuK. The L97D mutant appears to promote inner helix bending at the glycine gating hinge (Gly92) and the direction of the helix bend is parallel with the orientation of the linker between the pore and the gating ring. Its gating ring, however, is still in the $Ca^{2+}$-bound, closed conformation, which locks the intracellular gate closed at the end of the inner helix and prevents the concurrent movement of the TM2C with TM2B (**Figure 8C,D**). Consequently, the sharp turn between TM2B and 2C in the wild-type channel inner helix becomes straightened in the L97D mutant. These structural differences, along with the electrophysiological observation that the L97D mutant favors pore opening and has lower sensitivity to $Ca^{2+}$ deactivation, suggest that the L97D structure is in an intermediate state in which the pore inner helices are undergoing conformational changes towards opening but the channel's intracellular gate remains shut, deactivated by $Ca^{2+}$. With the gating ring constriction alleviated, we expect that the inner helix of the open pore undergoes a similar bending movement at the glycine hinge with TM2B and 2C moving concurrently. A working model for the open pore can therefore be generated by applying the inner helices bending observed at Gly92 of L97D mutant onto the wild-type pore and allowing TM2B and 2C to move as a rigid body (**Figure 11A**). In this open pore model, Leu117 is rotated away from the ion permeation pathway, yielding a larger entrance at the gate.

The proposed pore opening mechanics of GsuK is distinct from other K+ channels. The GsuK inner helix has two different hinge points in response to external stimuli, allowing the pore to

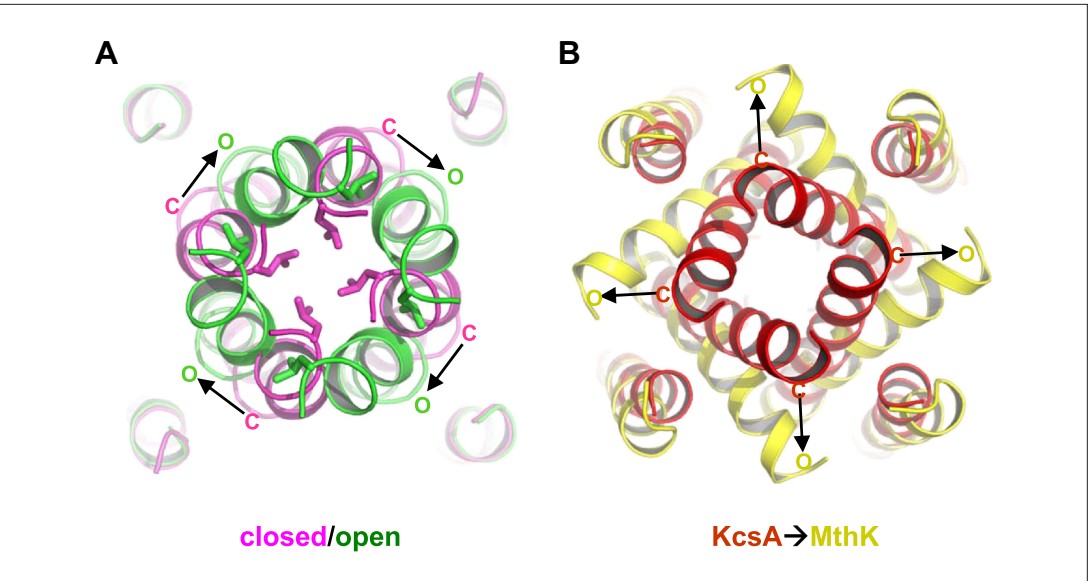

**Figure 11**. Proposed pore opening mechanics of GsuK. (**A**) Working model of the GsuK intracellular gate from closed (magenta) to open (green). Arrows indicate the direction of inner helix movement from closed to open state. Leu117 side chains are shown as sticks. (**B**) Pore opening mechanics of MthK. KcsA is used as the closed model for MthK.

adopt an open, closed, and $Ca^{2+}$-deactivated states. The GusK pore dilates open in a very different direction as compared to MthK or Kv channels (*Jiang et al., 2002b*; *Long et al., 2005*; *Ye et al., 2010*) (*Figure 11A,B*). This reversed dilation of pore opening between GsuK and MthK can be attributed to the differences in the relative orientation between the gating ring and the pore as shown in *Figure 7C*. As current structural studies of RCK-regulated gating suggest that the expansion of the gating ring upon activation is mechanically coupled to the pore opening (*Ye et al., 2006*; *Yuan et al., 2011*), then the inner helix movement in different directions is expected between GsuK and MthK.

Although protein samples were prepared in nominal $Ca^{2+}$-free conditions, the gating rings in all GsuK crystal structures are in a $Ca^{2+}$ bound, closed conformation, suggesting a relatively high $Ca^{2+}$ binding affinity in GsuK. The likely source of $Ca^{2+}$ is contamination of the chemicals used for protein purification and crystallization. Single channel recordings, however, showed a lower apparent affinity of $Ca^{2+}$ with a $K_{1/2}$ of about 200 μM. One possible explanation for this discrepancy is that the protein crystals were grown in a detergent environment whereas channel recording was performed in a lipid membrane. As lipids are known to be important for membrane protein stability, their presence may be necessary to stabilize the GsuK pore in an open state and, therefore, give rise to a lower efficacy of $Ca^{2+}$ deactivation. A possible region in GsuK for lipid interactions is the stretch of positively charged residues surrounding the closed gate at the carboxyl terminus of the inner helix, which could potentially interact with the phospholipid head groups and influence channel gating.

Certain resemblances between GsuK and Slo channels are noteworthy. Like GsuK, the majority of Slo2.1 and Slo3 channels have the same TVGFG signature sequence, and a study of mouse Slo3 showed a $K^+$ selectivity similar to that of GsuK (*Schreiber et al., 1998*). The Slo2 inner helix contains a conserved Pro at the position equivalent to the helix breaking Leu97 of GsuK, suggesting a similar inner helix break. $NAD^+$ has also been shown in a recent study to modulate $Na^+$ activation in Slo2 by binding at the same nucleotide site in RCK2 (*Tamsett et al., 2009*). Furthermore, the relative orientation between the transmembrane pore and the gating ring in Slo channels, and consequently pore opening mechanism, may have a closer resemblance to GsuK than MthK. One piece of evidence is that the acidic Asp90 residue in the S0–S1 loop of the BK channel transmembrane pore has been shown to interact with the Glu374/Glu399 $Mg^{2+}$ site from the neighboring subunits (*Yang et al., 2008*). Docking of the Kv channel pore onto the BK gating ring using MthK as a model would position such

interaction within the same subunit (*Wu et al., 2010*), but is otherwise possible if using GsuK as a structural model.

## Materials and methods

### Protein expression and purification

The *GsuK* gene from *Geobacter sulfurreducens* was initially subcloned into the pQE 70 vector using Sph I and Bgl II restriction sites, and contains a thrombin cleavage site between the C-terminal His-tag and the channel. The N-terminus of GsuK was modified for the expression of both the full-length channel and its intracellular subunit. The construct for full-length channel starts at residue Ala9 and contains five extra amino acids (MQRGS) at the N-terminus. The construct for the expression of the GsuK intracellular subunit starts at residue Tyr10 and contains two extra amino acids (MQ) at the N-terminus. Interestingly, although both constructs are very similar, the expression of the latter one starts from Met107 instead of the first Met, producing only the intracellular subunit. Despite the lack of the membrane-spanning segments, solubilization and purification of the GsuK intracellular subunit still requires the presence of detergent at a concentration above the CMC (critical micelle concentration).

Both constructs were expressed in *E.coli* BL21(DE3) cell lines by induction (at $A_{600} \sim 0.8$) with 0.4 mM isopropyl-β-D-thiogalactopyranoside (IPTG) at 37°C for 3–4 hr. Cells were harvested and lysed in a solution of 50 mM Tris–HCl, pH 8, 250 mM KCl and protease inhibitors including leupeptin, pepstatin, aprotinin and PMSF (Sigma-Aldrich, St. Louis, MO). Expressed proteins were then extracted from the cell lysate for 3 hr at room temperature in the above solution by adding 40 mM n-decyl-β-D-maltoside (DM). The detergent-solubilized proteins were loaded on a Talon $Co^{2+}$ affinity column (Clontech, Mountain View, CA) equilibrated with 50 mM Tris–HCl, pH 8.0, 250 mM KCl and 4 mM DM. In-gel digestion was performed by incubating the protein-bound $Co^{2+}$ resin with thrombin (1 unit per liter of bacterial culture) at 4°C overnight to remove the His-tag and released the proteins into the equilibration solution. After elution, proteins were further purified and buffer exchanged on a Superdex-200 (10/30) gel filtration column in a solution of 20 mM CHES, pH 9.0, 150 mM KSCN, 0.1 mg/ml *E. coli* polar lipid, 2 mM DTT and 4 mM DM for the full-length GsuK channel, and in a solution of 50 mM Tris–HCl, pH 8.0, 250 mM KCl, 2 mM DTT and 4 mM LDAO for the GsuK intracellular subunit. Both proteins elute at a position corresponding to the size of a tetramer.

### Crystallization and structure determination

Purified GsuK intracellular subunit was concentrated to approximately 8 mg/ml using an Amicon Ultra centrifugal filtration device (50 kDa MW cutoff) and crystallized at 20°C using the sitting drop vapor diffusion method by mixing equal volumes of concentrated protein and well solution containing 20–23% PEG3350, 120 mM KCl, 80 mM $NaNO_3$, 1% glycerol and 100 mM Bis-Tris propane, pH 8.5. The crystals were cryo-protected by slowly supplementing the crystallization drops with extra 20% PEG400 and flash frozen in liquid nitrogen.

Two mutations, E52A and Q77E or R, were introduced to the pore region of the full-length channel in order to obtain diffracting crystals. Both mutations did not have any observable effect on channel function as tested in single channel electrophysiology, and therefore this full-length channel is considered as wild-type in this study. The crystal quality was further improved by supplementing the buffer solutions with *E. coli* polar lipids during protein purification. The protein was purified in DM using the same procedure as described above and concentrated to approximately 6 mg/ml for crystallization at 20°C with well solution containing 13–18 % PEG3350, 250–500 mM KSCN, and 100 mM CHES, pH 9.0. The additional L97D mutation gave rise to better diffracting full-length channel crystals under the same crystallization condition. This L97D mutant was also used for co-crystallization with nucleotides where ADP or $NAD^+$ was added to the protein solution to a final concentration of 1 mM before crystallization trials. All full-length channel crystals were cryo-protected by slowly increasing the PEG3350 concentration in the crystallization drops to 20% followed by a supplement of 20% PEG400.

X-ray data were collected at the Advanced Photon Source (APS) Beamlines 19-ID and 21-ID, and at the Advanced Light Source (ALS) of the Lawrence Berkeley Laboratory (LBL) beamline 8.2.1. Data processing and scaling were performed using the HKL2000 software (*Otwinowski and Minor, 1997*). Crystals of the GsuK intracellular subunit are of space group *I222* with unit cell dimensions of a = 110.6 Å, b = 161.7 Å, c = 310.1 Å, and α = β = γ = 90°, and contains four subunits, which form a gating ring

in the asymmetric unit. The structure was determined by molecular replacement method using the open MthK gating ring (PDB ID: 1LNQ) as the search model followed by repeated cycles of model building with XtalView (*McRee, 1999*) and refinement with REFMAC (*Collaborative Computational Project, 1994*). The final model was refined to 3.0 Å with $R_{work}$ of 22.8% and $R_{free}$ of 26.2% (*Table 1*) and contained residues from 110 to 564 of each subunit. The full-length GsuK crystals are of space group *C2* and contain four subunits in an asymmetric unit. The four subunits do not belong to the same channel tetramer, but instead participate in the formation of two channel tetramers with their crystallographic twofold related partners. The structure was determined by molecular replacement method using half of the GsuK gating ring structure (two intracellular subunits) as the search model, followed by repeated cycles of model building in Coot (*Emsley and Cowtan, 2004*) and refinement with PHENIX (*Adams et al., 2010*). The final models for the wild-type channel and L97D mutant were refined to 3.7 Å and 2.6 Å, respectively. In the models of all full-length channel structures, two of the subunits contain residues 17 to 564 whereas the subdomains of the other two subunits (residues 262 to 349 and residues 481 to 564) are disordered. Detailed data collection and refinement statistics are listed in *Table 1*. All structure figures were generated in PyMOL (*DeLano, 2002*). The cavity space within the ion conduction pore was analyzed in HOLLOW (*Ho and Gruswitz, 2008*).

## Protein reconstitution and [86]Rb flux assay

The full-length channel proteins purified in DM was reconstituted into lipid vesicles composed of a 3:1 ratio of 1-palmitoyl-2-oleoyl-phosphatidylethanolamine (POPE) and 1-palmitoyl-2-oleoyl-phosphatidyl glycerol (POPG) (Avanti Polar Lipids, Alabaster, Al) as described (*Heginbotham et al., 1999*; *Alam et al., 2007*) using a dialysis solution containing 10 mM HEPES, pH 7.4 and 450 mM KCl. The reconstituted liposome samples were kept at −80°C in 100 µl aliquots. For [86]Rb flux assays, a protein/lipid ratio of 10 µg/mg was used in the reconstitution.

The [86]Rb flux assay was performed following the same procedures as described (*Heginbotham et al., 1998*). Liposomes were thawed and sonicated in a bath sonicator for 30 s before the assay. To remove extra-liposomal KCl, samples were passed through a pre-spun Sephadex G-50 fine gel filtration column (1.5 ml bed volume in a 5 ml disposable spin column) swollen in 450 mM Sorbitol and 10 mM HEPES, pH 7.4. Each tested nucleotide was added to a 30 µl aliquot of the liposomes collected after the buffer exchange step followed by the addition of 56 µl [86]Rb flux buffer (450 mM Sorbitol, 10 mM HEPES, pH 7.4, 50 µM KCl, and 5 µM [86]RbCl). The final concentration of the nucleotide in the reaction mixture is 1 mM. After 5 min, this reaction mixture was passed through another pre-spun gel filtration column as described above to eliminate extraliposomal [86]Rb. The final eluate was mixed with 10 ml scintillation cocktail and its radioactivity measured in a scintillation counter. The radioactivity of each sample was normalized against the control sample in which no nucleotide was added.

## Single channel recordings from giant liposome patch

Initial single channel recordings using lipid bilayers or giant liposome patching with vesicles reconstituted with the GsuK channel failed to detect channel activity. Low open probability and small single channel conductance can both contribute to the lack of channel activity in electrophysiological recording. Our recent study of the MthK channel demonstrated that mutations at Ala88 on the MthK inner helix can have dramatic effects on single channel conductance and open probability—replacing Ala88 with a larger hydrophobic residue such as Leu significantly reduces both whereas the opposite effect is seen with a negatively charged Asp residue (*Shi et al., 2011*). The equivalent residue in GsuK is Leu97, which we reasoned could potentially be the cause of low channel activity. Furthermore, $Ca^{2+}$ binding at the assembly interface of GsuK also implies a potential gating role. To this end, we introduced an L97D mutation to enhance the channel conductance and/or open probability and also performed single channel recordings in the presence and absence of $Ca^{2+}$. As the results showed, the $Ca^{2+}$ ion initially presented in the patching solutions is the main cause of low channel activity.

For single channel recordings, a protein/lipid ratio 0.5–2 µg/mg was used in the reconstitution. Giant liposome was obtained by air drying 2–3 µl of liposome sample on a clean cover slip overnight at 4°C followed by rehydration in bath solution at room temperature. Patch pipettes were pulled from Borosilicate glass (Harvard Apparatus, Holliston, MA) to a resistance of 8–12 MΩ upon filled with the pipette solution containing 150 mM NaCl (for recordings shown in *Figure 6B,C*) or KCl (for recordings shown in *Figure 6E,F*), 1 mM $CaCl_2$, and 10 mM HEPES, pH 7.4 buffered with KOH or NaOH. The standard bath solution contains 150 mM KCl, 10 mM HEPES, 1 mM EGTA, pH 7.4 buffered with KOH.

A giga seal (>10 GΩ) was obtained by gentle suction when the patch pipette attached to the giant liposome. To get a single layer of membrane in the patch, the pipette was pulled away from the giant liposome and the tip was exposed to air for 1–2 s. Membrane voltage was controlled and current recorded using an Axopatch 200B amplifier with a Digidata 1322A converter (Axon Instruments, Union City, CA). Currents were low-pass filtered at 1 kHz and sampled at 20 kHz. Only patches containing a single channel with $P_o$>0.85 in the absence of $Ca^{2+}$ and having its intracellular side facing the bath solution were used for further experiments. In the study of intracellular $Ca^{2+}$ deactivation, various concentrations of $Ca^{2+}$ were added to the bath solution. The free $Ca^{2+}$ concentration in the range of 0–100 µM was controlled by mixing 1 mM EGTA with an appropriate amount of $CaCl_2$ calculated using the software MAXCHELATOR (http://maxchelator.stanford.edu). No EGTA was added in the bath solution for $[Ca^{2+}]$ above 100 µM. Most recordings were performed with symmetrical KCl except for the selectivity measurements in which KCl in the pipette solution was replaced by 150 mM NaCl.

As the channel in liposome patch has its intracellular ligand binding gating ring facing the bath solution, the positive (outward) current is defined as the cation movement from the bath solution to the pipette. The presence of 1 mM $CaCl_2$ in the pipette solution (extracellular to the channel) has no effect on channel open probability both in negative and positive voltages, which rules out the possibility of a slow blockade of the pore by $Ca^{2+}$.

## Acknowledgements

We thank Nam Nguyen for discussion and critical review of the manuscript. We thank the beamline staff from the Advanced Photon Source (APS) and the Advanced Light Source (ALS) for assistance in data collection. Use of the Advanced Photon Source, an Office of Science User Facility operated for the U.S. Department of Energy (DOE) Office of Science by Argonne National Laboratory, was supported by the U.S. DOE under Contract No. DE-AC02-06CH11357. The atomic coordinates and structural factors have been deposited in the Protein Data Bank with the accession number 4GVL for the GsuK gating ring, 4GX5 for the wild-type GsuK, 4GX0 for the L97D mutant, 4GX1 for the ADP complex of L97D mutant and 4GX2 for the $NAD^+$ complex of L97D mutant.

## Additional information

### Funding

| Funder | Grant reference number | Author |
| --- | --- | --- |
| Howard Hughes Medical Institute | 012229 | Weizhong Zeng, Liping Chen, Youxing Jiang |
| Welch Foundation | I-1578 | David Bryant Sauer, Yeeling Lam |
| Natural Science Foundation of Zhejiang Province, China | R2100439 | Sheng Ye |
| National Institutes of Health | RO1 GM071621 | Chunguang Kong, Mehabaw Getahun Derebe |
| David and Lucile Packard Foundation | 2004-27671 | Youxing Jiang |
| Specialized Research Fund for the Doctoral Program of Higher Education, China | 20110101110122 | Sheng Ye |

The funders had no role in study design, data collection and interpretation, or the decision to submit the work for publication.

### Author contributions

CK, Conception and design, Acquisition of data, Analysis and interpretation of data; WZ, Acquisition of data, Analysis and interpretation of data; SY, Analysis and interpretation of data; LC, Acquisition of data, Analysis and interpretation of data; DBS, Drafting or revising the article; YL, Acquisition of data; MGD, Acquisition of data; YJ, Conception and design, Acquisition of data, Analysis and interpretation of data, Drafting or revising the article

## Additional files

### Major datasets

The following datasets were generated

| Author(s) | Year | Dataset title | Dataset ID and/or URL | Database, license, and accessibility information |
|---|---|---|---|---|
| Kong C, Zeng W, Ye S, Chen L, Sauer DB, Lam Y, Derebe MG, Jiang Y | 2012 | Atomic coordinates and structural factors of GsuK gating ring | Accession number 4GVL | Publicly available at the RCSB Protein Data Bank (http://www.rcsb.org/pdb/) |
| Kong C, Zeng W, Ye S, Chen L, Sauer DB, Lam Y, Derebe MG, Jiang Y | 2012 | Atomic coordinates and structural factors of the wild-type GsuK | Accession number 4GX5 | Publicly available at the RCSB Protein Data Bank (http://www.rcsb.org/pdb/) |
| Kong C, Zeng W, Ye S, Chen L, Sauer DB, Lam Y, Derebe MG, Jiang Y | 2012 | Atomic coordinates and structural factors of GsuK L97D mutant | Accession number 4GX0 | Publicly available at the RCSB Protein Data Bank (http://www.rcsb.org/pdb/) |
| Kong C, Zeng W, Ye S, Chen L, Sauer DB, Lam Y, Derebe MG, Jiang Y | 2012 | Atomic coordinates and structural factors of GsuK L97D mutant in complex with ADP | Accession number 4GX1 | Publicly available at the RCSB Protein Data Bank (http://www.rcsb.org/pdb/) |
| Kong C, Zeng W, Ye S, Chen L, Sauer DB, Lam Y, Derebe MG, Jiang Y | 2012 | Atomic coordinates and structural factors of GsuK L97D mutant in complex with NAD | Accession number 4GX2 | Publicly available at the RCSB Protein Data Bank (http://www.rcsb.org/pdb/) |

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
