## [Decision Letter]

Thank you for choosing to send your work entitled “Distinct Gating Mechanisms revealed by the structures of a multi-ligand gated K^+^ channel” for consideration at *eLife*. Your article has been evaluated by a Senior Editor and 3 reviewers, one of whom is a member of *eLife's* Board of Reviewing Editors.

The Reviewing Editor and the other reviewers discussed their comments before we reached this decision, and the Reviewing Editor has assembled the following comments based on the reviewers' reports. Our goal is to provide the essential revision requirements as a single set of instructions, so that you have a clear view of the revisions that are necessary for us to publish your work.

This paper describes the first structural and functions studies on a new bacterial K^+^ channel from *Geobacter sulfurreducens* called GsuK. GsuK is closely related the bacterial MthK channel and also related to the eukaryotic Slo family of K^+^ channels that includes Slo1 (BK), a Ca^2+^ activated K^+^ channel. However, unlike these channels, these authors show that GsuK is inhibited by Ca^2+^ and weakly up-regulated by ADP and NAD^+^. The authors obtained X-ray crystal structures of several forms of the channel including the intracellular gating ring formed from a tetramer of two tandemly linked RCK domains, and the nearly full-length channel that includes the transmembrane domains and the gating ring. While generally the structures are similar to the previously crystalized MthK channels, they reveal a number of very interesting differences and unique observations.

For example, the inner helix that lines the pore is “segmented”, containing a bend not just at the PVP site of Kv channels, but also at a second site more intracellular. This produces a second bundle crossing, and presumed gate, more intracellular than previously found. The structure of a mutant channel, L97D, suggests that the inner helices move in a different direction from MthK and Kv channels, perhaps because the gating ring is rotated relative to the transmembrane domains. The authors observe density (modeled as AMP) that likely corresponds to a nucleotide binding site, and they demonstrate apparent NAD^+^ and ADP-dependent transport by these channels using Rb fluxes. In addition, the authors observe Zn coordinated by two His and two Cys sidechains in the RCK domain, and another ion modeled as Ca. Because none of these ions or nucleotide were added during purification or crystallization, it would appear that each originated from the *E. coli* cytoplasm and/or culture medium and are very tightly bound to the protein. The inhibitory Ca^2+^ binding site is in a different place than the Ca^2+^ binding site of MthK and close to the site in BK that produces channel activation.

These results are clearly a major advance in our understanding of the molecular mechanism of ligand-regulated K^+^ channels. In particular, the differences in structure and mechanism with other channels in this family were unexpected, and could change the paradigm for how we think about the molecular mechanisms of gating in this family of channels. The results would also be of broad interest to people working on other ligand regulated enzymes. The work is well done and nicely presented. However, the authors' discussion and interpretation of gating mechanism is, in general, not well substantiated by the functional data that accompanies presentation of the structure, and this will require revision. The majority of these significant problems arise from over interpretation and over generalization between different channels. While they should be correctable without further experiments, a substantial rewrite will be required. A suitably revised paper would be an important contribution to the field. Comments that must be satisfactorily addressed:

1. The major weakness is an extreme interpretation of the data in terms of gating conformational changes in the membrane spanning pore domain. The structures of only presumed closed states are presented. Comparison with the structure of a mutant with altered calcium sensitivity and small conformational differences is used to argue for a gating conformational change at the cytoplasmic entrance. Much is made of this that goes beyond what is supported by the data. Some speculation and hypothesis generation along these lines is certainly appropriate, but the authors' assertions of details about the gating mechanisms are not supported by the data, beyond suggesting interesting hypotheses. The large amount of space devoted to gating mechanisms need to be pared down significantly and presented with more caution.

2. Another problem arises from the claimed similarities in gating between GusK and BK channels. There is a significant amount of literature arguing that BK channels do not have a significant cytoplasmic gate in the “bundle crossing” region, and that the activation “gate” is located closer to the selectivity filter. None of this work is mentioned. Taking these earlier studies on BK into account, along with the structure based interpretations on GsuK presented here, it seems more reasonable to claim that GsuK and Slo1 have different gating mechanisms.

3. Likewise the arguments for different calcium dependent and voltage dependent gates in BK based on interpretations of the GsuK full-length structure and the BK gating ring structure are groundless. As stated by the authors “In other word[s], voltage activation may control a different gate in the pore [...] allowing the channel to be independently regulated by both Ca^2+^ and voltage (Horrigan and Aldricxh, 2002).” This is the opposite of what Horrigan and Aldrich proposed. They proposed that Ca^2+^ and voltage independently act on the *same* gate. If there were two independent gates, then Ca^2+^ alone or voltage alone could never open the channel (but they do).

4. It would be helpful to see the time course data for the Rb flux experiments, as opposed to a single time point for each ligand, if these data are available, so that a reader might be able to compare the flux experiment time courses with those from other K channels.

5. If the currents in Figure 6B and C were recorded with 150 Na out and 150 K in, then the large inward current in these recordings would be carried entirely by Na. Most investigators would thus classify GsuK as a non-selective cation channel, and not a K^+^ channel, regardless of the selectivity filter sequence. So to refer to GsuK as a K^+^ channel is somewhat misleading.

6. The authors hypothesize that Ca is inhibiting the channel through the identified Ca binding site in the gating ring, because Ca added to the intracellular side of the channel decreases Po but not single-channel conductance. However, in Figure 6D, only inward currents are shown. If Ca is inhibiting through a slow blockade of the pore (i.e. like Ba), then one might expect this inhibition to show some voltage-dependence; this might be revealed if outward currents were analyzed as well. Alternatively, the authors' hypothesis might be supported through recordings of GsuK with a mutation at the Ca-binding site. Without any additional information, the argument that the Ca binding site in the RCK domain is the sole locus of Ca's inhibitory action is a weak one, and thus the authors should at least mention that other sites or mechanisms of inhibition cannot be ruled out.

7. Although a small increase in Po is shown with added nucleotide for Ca-inhibited channels (Figure 6E), it is also not clear whether this effect arises from the action of nucleotide at the binding site identified in the crystal, or from a non-specific action. This concern arises from the apparent tight-binding of nucleotide in the crystal and relatively weak interaction observed in the e-phys recordings. This issue might also be resolved through recordings of GsuK with a mutation at the nucleotide-binding site. In the absence of such data the conclusions should be toned down.

8. The section “Mechanisms of nucleotide activation” mainly describes nucleotide coordination, and so this section should probably be retitled “Mechanisms of nucleotide coordination”. It is not clear how one can discuss nucleotide activation of the channel when it is not clear from the single channel recordings that the channel is substantially activated by nucleotide, and the observed conformational changes with added nucleotide are so subtle. It is worth calculating a deltadeltaG from the increase in open probability with nucleotide (at low probability) and compare to a similar calculation for Ca^2+^ (will probably be a lower estimate). Perhaps ADP and NAD^+^ should be called modulators or regulators and not ligands.

9. The model in Figure 11 is confusing. It implies that ADP or NAD^+^ can only bind to the closed (Ca^2+^ inhibited) state. Since ADP and NAD^+^ are up-regulators, they probably bind with higher affinity to the open state. Just because there is no effect in the absence of Ca^2+^ does not mean that the regulators are not binding: the open probability is already near one.

---

## [Author Response]

*1. The major weakness is an extreme interpretation of the data in terms of gating conformational changes in the membrane spanning pore domain. The structures of only presumed closed states are presented. Comparison with the structure of a mutant with altered calcium sensitivity and small conformational differences is used to argue for a gating conformational change at the cytoplasmic entrance. Much is made of this that goes beyond what is supported by the data. Some speculation and hypothesis generation along these lines is certainly appropriate, but the authors' assertions of details about the gating mechanisms are not supported by the data, beyond suggesting interesting hypotheses. The large amount of space devoted to gating mechanisms need to be pared down significantly and presented with more caution*.

The major concerns from the reviewers center on the discussion about channel gating. We agree that certain discussions, especially about BK channels, are very speculative. The discussion about gating has been pared down significantly, making the paper more focused on data rather than speculation.

*2. Another problem arises from the claimed similarities in gating between GusK and BK channels. There is a significant amount of literature arguing that BK channels do not have a significant cytoplasmic gate in in the “bundle crossing” region, and that the activation “gate” is located closer to the selectivity filter. None of this work is mentioned. Taking these earlier studies on BK into account, along with the structure based interpretations on GsuK presented here, it seems more reasonable to claim that GsuK and Slo1 have different gating mechanisms*.

This concern as well as the following one is related to the first summary point raised by the reviewers. It is apparent that the discussion about BK channel gating in this paper is too speculative and therefore we removed this discussion in the revision.

*3. Likewise the arguments for different calcium dependent and voltage dependent gates in BK based on interpretations of the GsuK full-length structure and the BK gating ring structure are groundless. As stated by the authors “In other word[s], voltage activation may control a different gate in the pore [...] allowing the channel to be independently regulated by both Ca2+ and voltage (Horrigan and Aldrich, 2002).” This is the opposite of what Horrigan and Aldrich proposed. They proposed that Ca*^*2+*^
*and voltage independently act on the same gate. If there were two independent gates, then Ca*^*2+*^
*alone or voltage alone could never open the channel (but they do)*.

As mentioned in our response to point two, the discussion of BK channel gating may not be appropriate for this study and has been removed.

*4. It would be helpful to see the time course data for the Rb flux experiments, as opposed to a single time point for each ligand, if these data are available, so that a reader might be able to compare the flux experiment time courses with those from other K channels*.

We do not have the data for time-dependent Rb influx for each ligand. While the flux assays helped us identify potential nucleotide ligand in our initial study, they are too qualitative for kinetic analysis.

*5. If the currents in Figure 6B and C were recorded with 150 Na out and 150 K in, then the large inward current in these recordings would be carried entirely by Na. Most investigators would thus classify GsuK as a non-selective cation channel, and not a K*^*+*^
*channel, regardless of the selectivity filter sequence. So to refer to GsuK as a K*^*+*^
*channel is somewhat misleading*.

There appears to be a misunderstanding about the channel configuration in our study. In our liposome patch, the intracellular ligand binding gating ring is facing the bath solution, which is opposite to the conventional patch in whole-cell configuration. The negative current in our study is defined as the cation movement from the bath solution (with 150 mM K^+^) to the pipette (with 150 mM Na^+^). Therefore, the large negative current the reviewer mentioned here is actually carried by K^+^ (equivalent to outward current for the channel). Based on the reversal potential, the channel is about five-fold selective for K^+^. To avoid any confusion, we clarify the definition of the channel orientation and current directions in the methods section.

*6. The authors hypothesize that Ca is inhibiting the channel through the identified Ca binding site in the gating ring, because Ca added to the intracellular side of the channel decreases Po but not single-channel conductance. However, in Figure 6D, only inward currents are shown. If Ca is inhibiting through a slow blockade of the pore (i.e. like Ba), then one might expect this inhibition to show some voltage-dependence; this might be revealed if outward currents were analyzed as well. Alternatively, the authors' hypothesis might be supported through recordings of GsuK with a mutation at the Ca-binding site. Without any additional information, the argument that the Ca binding site in the RCK domain is the sole locus of Ca's inhibitory action is a weak one, and thus the authors should at least mention that other sites or mechanisms of inhibition cannot be ruled out*.

This concern is related to the same orientation issue raised in point 5. The negative current in Figure 6D is actually outward current with respect to the channel and we do not see the change in conductance with the presence of Ca^2+^ in the bath solution. In our liposome patch, the pipette solution (extracellular to the channel) actually contains 1mM CaCl2 and we did not see any effect on channel open probability both under negative and positive voltages. This also rules out the possibility of a slow blockade of the pore by Ca^2+^. Clarification of this point has been included in the revision.

[Editors' note: at the request of the editors, the authors subsequently changed the signs of single channel recordings and I-V curves in Figure 6 to conform to the standard convention.]

*7. Although a small increase in Po is shown with added nucleotide for Ca-inhibited channels (Figure 6E), it is also not clear whether this effect arises from the action of nucleotide at the binding site identified in the crystal, or from a non-specific action. This concern arises from the apparent tight-binding of nucleotide in the crystal and relatively weak interaction observed in the e-phys recordings. This issue might also be resolved through recordings of GsuK with a mutation at the nucleotide-binding site. In the absence of such data the conclusions should be toned down*.

The nucleotide binding site in a subset of RCK domains is highly conserved in sequence and structure. NAD and ADP have also been observed in several structures of RCK domains from K^+^ transporters and are classified as the ligand for the transporters. As we discussed in the paper, Ca^2+^ appear to have a major effect on channel gating. Although not as profound, the ADP and NAD do have clear and specific effect on channel gating, which is unlikely to arise from non-specific action. The low efficacy of the nucleotide ligand does not necessarily reflect weak binding. Further functional study on GsuK gating, such as mutations at the nucleotide binding site, will be pursued, but is not within the scope of this study, which is more focused on structural aspects of the channel.

*8. The section “Mechanisms of nucleotide activation” mainly describes nucleotide coordination, and so this section should probably be retitled “Mechanisms of nucleotide coordination”. It is not clear how one can discuss nucleotide activation of the channel when it is not clear from the single channel recordings that the channel is substantially activated by nucleotide, and the observed conformational changes with added nucleotide are so subtle. It is worth calculating a deltadeltaG from the increase in open probability with nucleotide (at low probability) and compare to a similar calculation for Ca*^*2+*^
*(will probably be a lower estimate). Perhaps ADP and NAD*^*+*^
*should be called modulators or regulators and not ligands*.

This is related to point seven. As mentioned in our response to point seven, we observed specific NAD and ADP binding in the structure and their effect on channel open probability (despite being less profound than Ca^2+^). NAD and ADP were observed in a subset of RCK domains that contain the conserved nucleotide-binding motif and were considered as the ligand for the RCK-containing K^+^ transporter. We feel that it would be more appropriate to consider the nucleotide as a ligand for GsuK instead of a modulator or regulator. The characterization of GsuK ligand gating including the mutation of nucleotide binding site as well as the quantification of ligand effect will be pursued in our future studies. We have changed the subtitle for this section to “ADP and NAD^+^ binding in GsuK”.

*9. The model in Figure 11 is confusing. It implies that ADP or NAD*^*+*^
*can only bind to the closed (Ca*^*2+*^
*inhibited) state. Since ADP and NAD*^*+*^
*are up-regulators, they probably bind with higher affinity to the open state. Just because there is no effect in the absence of Ca*^*2+*^
*does not mean that the regulators are not binding: the open probability is already near one*.

Figure 11 does not intend to imply that ADP or NAD^+^ can only bind to the closed (Ca^2+^ inhibited) state. What we intend to show is that ADP and NAD^+^ binding can increase channel opening in the presence of Ca^2+^. We actually agree with the reviewers that ADP and NAD^+^ probably have higher affinity to the open state channel. As the working model in Figure 11 merely serves as a summary of some possible states discussed in the paper and does not add much new information to the paper, we removed it in the revision to avoid any confusion.